# How Does Living in Temporary Accommodation and the COVID-19 Pandemic Impact under 5s’ Healthcare Access and Health Outcomes? A Qualitative Study of Key Professionals in a Socially and Ethnically Diverse and Deprived Area of London

**DOI:** 10.3390/ijerph20021300

**Published:** 2023-01-11

**Authors:** Diana Margot Rosenthal, Antoinette Schoenthaler, Michelle Heys, Marcella Ucci, Andrew Hayward, Ashlee Teakle, Monica Lakhanpaul, Celine Lewis

**Affiliations:** 1UCL Population, Policy and Practice Research and Teaching Department, UCL Great Ormond Street Institute of Child Health, University College London, London WC1N 1EH, UK; 2UCL Collaborative Centre for Inclusion Health, University College London, London WC1E 7HB, UK; 3Center for Healthful Behavior Change, Institute for Excellence in Health Equity, NYU Langone Health, New York, NY 10016, USA; 4Specialist Children and Young People’s Services, East London NHS Foundation Trust, London E15 4PT, UK; 5UCL Institute for Environmental Design and Engineering, The Bartlett School of Environment, Energy and Resources, University College London, London WC1H 0NN, UK; 6UCL Institute of Epidemiology and Health Care, University College London, London WC1E 7HB, UK; 7Public Health, London Borough of Newham, London E16 2QU, UK; 8Community Paediatrics, Whittington Health NHS, London N19 5NF, UK; 9North Thames Genomic Laboratory Hub, Great Ormond Street Hospital, London WC1N 3BH, UK

**Keywords:** child homelessness, family homelessness, temporary accommodation, social determinants of health, COVID-19, qualitative, inclusion health, policy and practice, health inequities

## Abstract

Background: Children < 5 years living in temporary accommodation (U5TA) are vulnerable to poor health outcomes. Few qualitative studies have examined service provider perspectives in family homelessness; none have focused on U5TA with a cross-sector approach. This study explored professionals’ perspectives of the barriers and facilitators, including pandemic-related challenges, experienced by U5TA in accessing healthcare and optimising health outcomes, and their experiences in delivering services. Methods: Sixteen semi-structured online interviews were conducted. Professionals working in Newham (London) with U5TA families were recruited from non-profit organisations, the health sector, and Local Authority. A thematic analysis was conducted. Findings: Professionals described barriers including poor parental mental health; unsuitable housing; no social support; mistrust of services; immigration administration; and financial insecurity. Digital poverty, language discordance, and the inability to register and track U5TA made them even less visible to services. Professionals tried to mitigate barriers with improved communication, and through community facilitators. Adverse pandemic effects on U5TA health included delay and regression in developmental milestones and behaviours. In-person services were reduced, exacerbating pre-existing barriers. Interpretation: COVID-19 further reduced the ability of professionals to deliver care to U5TA and significantly impacted the lives of U5TA with potential life-long risks. Innovative and tailored cross-sector strategies are needed, including co-production of public health services and policies focusing on early development, mental health support, employment training, and opportunities for parents/carers.

## 1. Introduction

Under 5s experiencing homelessness in temporary accommodation (U5TA) are particularly vulnerable to poor health outcomes and poor healthcare service access in comparison to the general population [1,2] due to their transient circumstances and lower visibility, unlike other groups experiencing homelessness, such as rough sleepers [3]. 

On 11th March 2020, COVID-19 was declared a global pandemic, and on 23rd March 2020, the first of many pandemic lockdown measures was implemented in England with the intention of reducing transmission of the virus [4,5]. Although the general population was affected by these measures, those already considered vulnerable were even more so. The first five years of life is a critical development period; thus, the longevity and periodicity of the COVID-19 pandemic will likely have short-term impacts on under 5s that extend across most of their infant lives, with some children being born into the pandemic. On top of that, long-term impacts on children including adverse childhood experiences will likely carry over to adulthood (e.g., chronic health conditions, education, employment, and relationships) [3,6,7].

Living in temporary accommodation (TA) is a form of homelessness [8]. According to the Office of National Statistics, “Temporary accommodation may be provided while an assessment decision is being made or while homeless households are waiting for longer-term accommodation [8].” TA types are heterogeneous: some are intended for short-term (days/weeks), while others are intended for longer-term stays. During the pandemic, the number of families in TA across England significantly increased: approximately 253,000 people in England were experiencing homelessness in TA—the highest figure in fourteen years [9]. In 2021, the London Borough of Newham (LBN) reported the highest local rate of homelessness in England (1 in 22 people) [10]. In 2020 and 2021, 1 in 11 children were living in TA and 1 in 2 were living in poverty in LBN [11,12,13]. By the end of December 2020, 5664 Newham households were in TA, an overall 7.3% increase from 2019 (pre-pandemic), with an 11.3% increase in nightly paid, privately managed accommodation, or self-contained TA [11]. 

Previous mixed-methods studies have shown that U5TA experience numerous multi-level, interrelated barriers to accessing healthcare services and optimising health outcomes, even before the pandemic [2,14,15]. Few qualitative studies have been conducted to examine the professional perspective in the field of family homelessness pre-pandemic. Results from these studies have identified several professional barriers including inadequate medical education and training, poor consultation style, practice factors (e.g., workload/time), patients with multiple co-morbidities and high risk of non-compliance, and policies that hinder the provision of primary care services and social services communication thereby preventing access [16,17]. However, these studies did not discuss U5TA explicitly nor utilise a cross-sector approach (i.e., beyond health professionals) including all domains of access (e.g., affordability, accessibility, availability, accommodation, and acceptability) [18].

Therefore, the objectives of this study [19] were to (1) qualitatively explore what key professionals perceive as the main barriers and facilitators for U5TA in accessing healthcare services and optimising health outcomes and (2) elicit the professionals’ experience when providing services to this population before and during the COVID-19 pandemic (COVID). 

## 2. Methods

Ethics approval was obtained by the UCL Research Ethics Committee (ID number: 15097-001) and UCL Data Protection Office registration (no. Z6364106/2019/03/157). 

### 2.1. Participant Sample and Recruitment

To achieve maximum variation in the sample, a broad range of professionals who work with U5TA and their families were invited to participate in the interview study. The inclusion and exclusion criteria were as follows: ○Inclusion Criteria:
▪Professional who is currently working and/or has worked with U5TA and their families either directly or indirectly, i.e., does not interact with families but is still responsible for them (e.g., someone in the local authority or higher-up administration in a non-profit, services, management, etc.).▪Currently working in the London Borough of Newham (LBN) at the time of the interview.▪Come from one of the following stakeholder groups: Health Visitor (HV), Health Professional (HP), Non-profit Organisation (NP), and the Local Authority of LBN (LA).▪Professionals could be from any department, specialities, and/or field as long as they meet the other criteria.
○Exclusion Criteria:
▪Professional who has not worked with U5TA and their families in any capacity.▪Not currently working in LBN.

Potential participants were purposively sampled [20]. In addition, interview participants were asked to identify other potential participants (snowball sampling) [21]. Professionals were recruited between October 2020 and December 2021 through health visiting services, voluntary organisations, East London NHS Foundation Trust, and the LBN Public Health Team. Potential participants were emailed and sent a participant information sheet, and if interested, they were asked to reply via email with any questions they had about the study and/or to schedule the interview. A total of 61 professionals were contacted, of which 20 responses of interest were received. Of those, three were health visitors and unreachable after receiving the consent form and attempting to schedule the interview, and one professional was interviewed but later excluded after we determined they did not meet the inclusion criteria. Therefore, 16 interviews (26.2%) were included in the final analysis. Informed consent (either written or provided verbally and recorded) was obtained before the interview. Recruitment, interviews, and data analysis continued until thematic saturation was reached and no new codes emerged, which was judged at 14th–16th interviews [22].

### 2.2. Study Design

One-to-one semi-structured interviews were conducted on the videoconference platform Zoom [23], and a short demographic questionnaire (6 questions) was administered. Conducting interviews in this way has been shown to be effective [24], and studies have shown high satisfaction with this platform among health professionals for research purposes [25]. 

A thematic codebook approach was chosen to analyse the data [26], employing methods of organising the data from framework analysis [27] to facilitate comparisons across different professional participant groups for the presence or absence of themes [28]. Both inductive and deductive coding were included, including theory-driven and data-driven codes [29,30]. An adapted socio-ecological model (SEM) [2,31] was used as a guiding theoretical framework.

#### Topic Guide

An interview topic guide was developed based on previous findings of a larger multi-phase mixed-methods project [2,3,14,32] and by the lead author (DMR). A draft guide was reviewed and commented on by CL, followed by other co-authors. The topic guide covered the following areas: (1) definitions of homelessness (personally and professionally); (2) challenges when working with the U5TA population; (3) perceptions of U5TA challenges to accessing health services and optimising health outcomes; (4) impact of the pandemic on families and their services; and (5) recommendations to address any of the challenges discussed. The first few interviews (1 per professional group) acted as pilots, after which the interview guide wording was refined, and the order of questions was revised (Appendix A).

### 2.3. Data Analysis

Interviews were digitally recorded with audio transcription and then cross-checked for accuracy. All interviews were anonymised and given codes, e.g., HV1. An initial codebook was drafted using a priori deductive codes (from theory, topic guides, research questions, and previous findings results) [2,14,33,34,35] and inductive codes (from data familiarisation). The codes were labelled and defined by a description, inclusion and exclusion criteria, and origin, with an example quote [29]. To support rigour and replicability, three interviews were co-coded by CL and DMR independently to test the reliability of codes. After each transcript had been coded, CL and DMR compared coding. Where there were differences, these were discussed until consensus was reached and a revised codebook was created. A high level of agreement was reached using the fourth version of the codebook, and therefore, this codebook was used to code the remainder of the transcripts (Appendix A). In the final stage of the analysis, the data were ‘charted’ into three separate charts for each thematic heading of interest, i.e., individual/family, community, and systems level. Each chart was indexed (X-axis = sub-themes; Y-axis = professional groups) [28,30] using the matrices function in NVivo(v.12) [36]. This enabled within-case analysis of each participant to explore summaries of each participant’s view or between-case analysis to explore a specific theme across the various participants. An example matrix is provided in the Appendix A. 

## 3. Results

### 3.1. Professional Characteristics

Sixteen interviews were conducted (Med = 28 min; range = 14–41 min). Table 1 shows participant characteristics. Professionals included seven Health Visitors [HV], four Health Professionals [HP] including a General Practitioner (GP), therapist, dietician, and nurse, two Non-Profits [NP], and three Local Authority [LA] (e.g., public health consultant, social workers). At the time, professionals had worked in their current position in LBN for a range of 3 months to 21 years (Med = 2.0; SD = 5.1). All professionals were currently working in LBN, but some had previously worked in other boroughs, especially the HVs (Table 1).

#### Role Types and Services Provided

Each professional described the broad spectrum of services they provided in their roles, from frontline workers to research. For example, one consultant’s role involved mental health, housing, and employment support. Another liaised with the Home Office on behalf of families regarding their immigration status and with Housing, GPs, and schools to support families with no recourse to public funds (NRPF) as they regulate their status in the UK. Four HVs were specialist HVs working with children with special educational needs and/or disabilities (SEND). HVs described various roles and services, including needs and development assessment, toilet training, and family support. NPs said their roles were “…to support these families and by advising them on their housing options and ensuring that we assist to challenge bad housing.”(NP2)

### 3.2. Barriers and Facilitators 

Barriers and facilitators experienced by professionals and their perception of those experienced by U5TA families were organised using thematic headings taken from an adapted SEM [2], namely individual/family, community, and systems level (Figure 1). Within these three headings, data were broken down into meaningful sub-headings (e.g., mental health, cultural influences), facilitating comparison across the various professionals. Many of the barriers and facilitators discussed by participants existed before the pandemic. Others were exacerbated by (barriers) or were in response to (facilitators) the pandemic.

#### 3.2.1. Individual/Family Level

Professionals identified several individual/family-level barriers for U5TA families in optimising health outcomes and accessing healthcare. Many of these barriers mirrored their challenges in service provision to this population. The most frequently cited barrier was poor mental health, followed by language, cultural influences, immigration status, financial insecurity, and lack of trust in health services. In contrast, professionals only mentioned a few facilitators, including good communication skills and building trusting relations.

##### I. Poor Mental Health

Most interviewees discussed parental mental health as a barrier to U5TA maintaining healthy outcomes. These issues included anxiety, depression, and uncertainty due to their transient circumstances. HV2 discussed how poor parental mental health flows downstream, starting from the insecurity and unsuitability of the TA to the parent’s mental health, which then impacts the child.

*“…Well, [parents] cannot look after their children. So definitely, it’s like, it flows down, you know, so that is, I mean, that social effects is no good for the children that we work with. I think things that might make it easier for families to engage, you [know], to have good accommodation that is number one. If you have a roof over your head, if you are comfortable, if your mind is settled, definitely, we have better health outcomes for these families, but because you know they are depressed, a lot of anxiety…”* (HV2)

Multiple HVs and some HPs commented on how poor parental mental health affects positive parenting and subsequently, child health and development. HP3 discussed the availability of appointments and how long waitlists for speech and language therapy caused parents to feel helpless that they could not get the care they needed for their children during the most critical development period. Professionals echoed a similar sentiment of helplessness when they saw that U5TA families had so many competing priorities, preventing them from uptake of health services or following through with prescribed recommendations.

*“…then it stops them accessing everything you know. They’re so concerned about their housing that it just ripples out from there, just affects everything, so it really kind of almost makes me a bit redundant because I can’t help.”* (HP4)

HP4 put herself in their shoes and said she wanted to “wave a magic wand and make it all better” when a mother broke down in tears in front of her.

##### II. Language and Interpreter Challenges

All professional groups discussed that U5TA families face bi-directional language and cultural barriers. These included interpreters not being provided at GP surgeries or on automated calls and immunisation information not readily available in their language and/or a culturally acceptable format. This resulted in families reportedly accessing Accident and Emergency departmental services (A&E) out of convenience for routine clinical care or when a health issue became more severe because the parent could not communicate their children’s needs at the GP surgery. Professionals were notably surprised to report their own experience and families’ experiences of interpreters not provided through their service, given that LBN has more than a hundred spoken languages and rich ethnic diversity. However, they described it as “just another layer of struggle”(HP4) for U5TA families, noting that many essential health websites lacked translations of registration, immunisation information, and more for ESL/non-English speakers: 

*“So much information on the NHS website; they are like in British English. So, for people that do not have English as their first language, then it’s a big barrier because they don’t understand what you’re saying. Our areas as identified, you know, like a language barrier [in] homelessness is a key, key issue.”* (HV2)

HVs and NPs had their own challenges communicating with families. HVs reported the difficulties and “extra” pressures of arranging interpreters for families whose first language was not English and the time spent “…trying to kind of be able to communicate with families to find out what support they required and how we can support them best.” (HV7)

Language barriers were mitigated when professionals utilised an interpreter service when available called The Language Shop to accommodate families: 

*“…I think they’re all [language, cultural differences] definitely barriers, but I think that we always try to sort of think of ways around them like we always have access to [the] language line. So, we always have interpreters that we can use over the phone, which makes a really big difference.”* (NP1)

Therefore, these barriers did not mean there were zero interpreters but rather problems accessing an interpreter when needed to access the health service efficiently.

##### III. Cultural Differences and Influences

Professionals felt that cultural barriers impacted the uptake of immunisations. HV3 gave an example of Muslim parents’ reluctance to immunise their children if the ingredients in the vaccine contained pork or gelatine. HPs often neglected to communicate the alternatives (possibly due to language barriers), which resulted in mistrust between parents and HPs. Other cultural barriers included ‘culture shock’, e.g., LA1 described a client who was moved from LBN to Essex and felt like “an alien”, leading to feelings of isolation and unsettledness, which was interrelated with the community-level barriers:

*“…[One] Mum said to me today: ‘I went to the local shops’ because she was looking for some shops to buy her own cultural fruit because she’s African, and she said, ‘I couldn’t find those shops.’ She says she ‘feels like an alien down the street’ because she’s not… seeing the people that resemble her ethnicity, and so yeah, I understand what she means about the alien analogy."* (LA1)

One theme that emerged was how the advice provided by professionals might conflict with cultural practices, e.g., around the use of pillows for a newborn baby, as described by one of the HVs: 

*"For instance, you might tell them that for a newborn child, they don’t need to put them on a pillow or something, but in terms of some of the culture, they prefer to put the babies on the pillow to [give] support. They feel that otherwise, their head is not going to be the way they want it. They want the baby to lie still in one place and all that. So those are some things, and if you don’t understand the culture, they might see it as you’re being disrespectful or something.”* (HV5)

This HV also had concerns about whether health information was conveyed appropriately by interpreters, particularly if the interpreter came from the same background and culture as the family:

*“…sometimes language barrier also makes it very difficult to give them the information that you feel that they need, because sometimes you might even go [through an] interpreter, but then again, sometimes the interpreter might be biased because they are the same like this. They speak the same language, so sometimes what you’re telling them…you don’t know what they say to them.”* (HV5)

Professionals were aware of cultural sensitivity; it was not uncommon that families challenged professionals on their advice when it was not the cultural norm, especially among multi-generational households where different family members were making the healthcare decisions. 

*“In some families where they live in the hierarchical setup, the in-laws or the grandparents have a real influence on child rearing and development. So, in terms of seeking or adopting public health advice around weaning, for example, health, nutrition that [are] influenced by the practices of maybe the mother-in-law, that sister-in-law or the grandparents in terms of how a new mom would feed. So, you’ve got that conflicting advice, and you’ve also got some of its own things around cultural practices of what they did, culturally, which, as far as they’re concerned: ‘When I looked after you, you grew up fine. You’re fine’…”* (HV3)

Some specialist HVs also discussed these barriers relating to the child’s diagnosis, where parents may be reluctant to accept the diagnosis, especially those involving disability. Another barrier cited was families’ lack of knowledge of child protection systems (i.e., social care, policies) and how these work even with the aid of an interpreter because *“…it’s not within the normal culture, the normal understanding”*. (HV1)

##### IV. Immigration Status and Financial Insecurity

HVs, HPs, and LAs discussed immigration status as a general barrier to accessing health services. HV6 highlighted that immigrant families might not understand how to access services or how the UK healthcare system works. Many HVs perceived that families sometimes do not report their immigration status and “*stay under the radar*”(HP5) for fear of being reported to social services or deportation. 

Furthermore, professionals reported the pandemic had a significant impact on U5TA, primarily among those with NRPF status, making them even more vulnerable than they had already been pre-pandemic. This occurred for various reasons, including the cessation of new applications to the Home Office, the closure of many charities and other venues for getting support, and reduced ‘cash-in-hand’ work. 

*“…in terms of the deprivation that a lot of these families experience who do not have recourse to public funds because of their immigration issues, they are families who have struggled the worst, I would say, during the pandemic…families were stuck…didn’t have any status at the time, who were very much invisible, but were working for cash-in-hand, so they were maintaining their families. But the pandemic happened, and all the work stopped. Therefore, they weren’t able to pay their rent, so, therefore, they were coming through the front doors of social care, and these are the families that we’ve been working quite long and hard with since the beginning of COVID.”* (LA1)

According to professionals, these families were more likely to be put in hotel rooms where they could not cook traditional delicacies or healthy meals as recommended by HVs and HPs. Food insecurity was perceived as a prevalent issue as families could not afford three complete meals a day. Many families also relied on school meals and foodbank vouchers from HVs, described as a “*saving grace*”(HV3); however, many of these venues were closed during lockdowns. Financial insecurity also meant that families could not afford transport to access services based on the geographical location of their accommodation. Transientness was a barrier to schooling and employment opportunities for parents/carers, often resulting in financial insecurity.

U5TA families with NRPF (or insecure immigration) status had a secondary effect on professionals: families would sometimes disappear before their next health visit, and it was unclear whether they had returned to their country of origin. Rising financial insecurity and child poverty during the pandemic also led to an increased prevalence of homelessness and workload for professionals, especially NPs and LAs.

##### V. Mistrust and Communication Skills

Gaining the trust of U5TA families was of utmost importance. Community ties and developing trusting relationships with a particular ethnic and/or cultural group enabled professionals to deliver services to children within the community. 

*“…some people don’t believe in immunisation…and you tell them about immunisation, [and] it’s not something that they do, or they want to do…when I was in the Jewish community…Immunisation you have to literally [have to] win their trust before they can immunise their children, and if one person says to them, this nurse is good, you can let her immunise your child, yes, the whole of them will come, but if something negative, nobody will come to you, so you also need to win their trust.”* (HV5)

Families were reportedly “*a bit wary of professionals*”(HP5), suspicious of mainstream services, and felt like they had been failed by services (i.e., health, social, and housing).

*“I think the work that we have to do is about gaining their confidence in using our services. It may be that you’re dealing with a client that has been moved 20 times, so by the time that they see us, it might be that their faith in services is completely eroded because they just have kind of been left behind. They’ve been unseen, unknown, unnoticed….”* (HV4)

The general perception that families mistrusted healthcare providers, services, and the overall system appeared challenging for all professionals except NPs. Some HVs noted that their role was to earn back trust and help families rebuild their confidence in order to access services. 

Trust was also influenced by the professionals’ communication skills. Good communication skills were emphasised with a nod to shared decision making [37].

*" I think it’s more about understanding because health visiting services, in particular, is not something that’s universal, universally available across the world, so I think it’s kind of breaking the barriers in terms of understanding what our role is and what we can bring to that family. So, I think this is where [the] kind of our communication skills are important, you know, and it’s about utilising interpreters so that we can be clear to parents, so that…we know that they can understand what we’re saying to them and then they can be involved in the discussion as well."* (HV4)

HVs discussed keeping U5TA families engaged with services by contacting them about missed and follow-up appointments. HP5 explained how she was lenient about missed appointments when a family would be dropped from her service if they did not attend two consecutive appointments. Many professionals discussed “going the extra mile”(NP1) for U5TA families. NP1 gave an example of a mother who could not get her child’s inhalers because there was no interpreter at the GP to understand what she needed, and the child ended up in A&E because of asthma attacks. NP1 acted as a facilitator by intervening on the families’ behalf by identifying and communicating the actual problem as a logistic systems-level issue, not an issue of the parent’s neglect. 

#### 3.2.2. Community Level

The main barriers frequently highlighted were safety/suitability, overcrowding/shared facilities, lack of stimulation and outdoor space, dampness/mould, pests/vermin, and lack of social support/community presence.

##### I. Safety and Suitability

Professionals were asked what 1) they thought about the quality of TA and 2) if that affected U5TA health. The most common responses to 1) were “*unsuitable*,” “*poor quality*,” and “*not fit for purpose*”, and 2) “*100% percent*” and “*absolutely*”, respectively. Professionals argued that the TA environment was unsuitable for the needs of families.

*“…I don’t think I’ve been to any temporary accommodation, that is, you know, that is conducive for families. I don’t think I’ve been to any, to be honest with you.”* (HV2)

NP1 described that quality ranges by type of TA and suggested that LBN provides poor-quality accommodation.

All professional groups discussed the general safety of TA properties with a lack of security and safety provisions. There were multiple reports of unsafe, steep stairs which were not infant/childproofed or accessible. 

*“There were safety issues as well…one family was in a converted pub, and they had two rooms…there was a safety aspect in terms of the parents had to be with the children all the time. They had to go down the corridor and downstairs to do the washing up in a tiny sink. There were no kitchen facilities, and yeah, it was really, really tough for them…there was sort of like bunk beds in one room, so there wouldn’t have been anywhere for the children to play. The access to the building was a fire escape…[but] I think mom had a buggy [and how] she just used to struggle going up and down with the buggy, as well as the other children…”* (HP5)

TA with bunkbeds was assigned to families but was particularly unsuitable for U5TA with SEND. Unsuitable housing also included wheelchair users placed on the third floor of a building without a lift. HP5 and some HVs reported hazards to the landlord, LA, and local MP (Member of Parliament).

Professionals discussed how some U5TA tenants had experienced domestic abuse and perceived they did not feel safe in TA when sharing tenancy with individuals, especially those who struggled with drug addiction or were recently released from prison. In addition to reporting concern for the safety of U5TA, professionals feared for their own safety when delivering their service onsite:

*“It’s a B&B…but [I] think it’s now used mainly for our temporary accommodation. It’s a very big building, but even when you are going in as a professional, you are scared because the lighting in the building is not right…we see people loitering around smoking, you know, so you, yourself as a professional, you feel all scared and, you know, on top [of] homeless families that have young children, they’re not able to come out because it’s not safe for them…”* (HV2)

##### II. Overcrowding and Shared Facilities

Most professionals described overcrowding and shared facilities (e.g., bathrooms, kitchens) as barriers in the housing environment, including four or five family members living and sleeping in the same room. Whole families sharing one room supposedly caused more exposure and risk to infections, including COVID, especially during lockdown. 

*“A lot of the families I work with tend to share one very large room that sleeps about three, that sleeps a family of five, which I don’t think is very healthy because when one child has any sort of [infection], I’ve got a family of five at that moment where two of the children are three and a half and five. And obviously, they’re mixing at school. There are infections there in infants…So, when one child comes home with a cold, everybody will get it because they’re all sleeping in that room, even though they have access to separate kitchens and bathroom. I don’t think it’s fair. I don’t think it’s healthy or appropriate for parents to sleep in the same room as their children.”* (LA1)

Furthermore, some HVs said families were sleeping in the same bed, including newborns and infants, increasing the risk of Sudden Infant Death Syndrome and mental health difficulties. HPs worked with Scope, a sleep support charity, and a parents’ group in LBN called SENDIASS to mitigate the impact that housing had on children’s sleep.

Professionals were frustrated about the unsuitable housing environment because families could not follow their prescribed recommendations (e.g., therapies, diets). One HP described adapting her dietary advice for families who only had access to a microwave. Professionals described SEND as a sub-group of U5TA that is often missed from the conversation but perceivably experienced more significant challenges from the lack of space to store medical equipment (e.g., oxygen tanks) or perform prescribed physiotherapy, making their condition worse. 

*“I work with children with additional needs. There is no space for them to keep their medical equipment. So that means they can’t have the therapy or exercise that I prescribed by the therapist. They cannot be done because there is no place in the first instance to put the equipment.” *(HV2)

##### III. Lack of Stimulation and Outdoor Space

In addition to indoors, professionals described the lack of outdoor space and access, which meant that U5TA did not have a place to explore, play, run, or stimulate the senses, all of which are vital to the development of fine and gross motor skills. 

*“Children grow by, you know, play and stimulation, but in their case, they are limited; they can’t do all that. There is no garden for them to play, so they are confined to the four corners of the room, thereby making their health condition worse.” *(HV2)

Whilst participants acknowledged that even pre-pandemic U5TA were “*quite delayed in their speech*” due to lack of stimulation in the TA environment and sometimes “*were under percentiles for their age…*” (HP5), the pandemic further exacerbated individual health outcomes such as delay and regression in developmental milestones and behaviours including toileting, feeding skills, emotional regulation, and social-communication skills. 

*“But then in terms of their milestones, I think I have seen in terms of speech development…there’s been an increase in the amount of throwback I had to do in speech delay. And I think that’s definitely tied into COVID and that there are no children centres, so it’s stimulation issues, so it’s kind of lack of stimulation that could be the cause of it. But then, also the fact that there aren’t any facilities that are available in the Community for children to access, play and explore [or] school to help improve those skills [which] are lacking."* (HV4)

##### IV. Pests, Vermin, Dampness, and Mould 

All professional groups reported mould and dampness in most TA in LBN. Dampness and mould were reportedly the cause of skin irritations, respiratory illnesses, allergies, and exacerbation of asthma attacks.

*“Another major one is like infection. When they are in a more dirty environment, it affects their health, especially children with asthma. When they live in an environment that is damp with mould, it increases, you know, the rate at which they wheeze. It affects the respiratory system, and then they start having to go to Accident and Emergency all the time…” *(HV2)

Professionals reported the dangerous combination of having pests and vermin with a mobile infant or child in the early years and the impact this could have on their health:

*“ I would say breathing, definitely respiratory difficulties, which then leads onto triggering other things like skin irritations, especially when you’ve got pests, and there’s droppings around the properties. [Be]cause children, especially under-fives, they will crawl, they will walk, they will pick things up and put them in their mouths and so, there’s quite a lot that I would say would affect them and that could be the worst-case scenario that affects their development.”* (LA1)

##### V. Social Support and Community Presence

Charities, children’s centres, and community centres provided links for professionals to access U5TA and for families to access healthcare services, which was crucial and bi-directionally beneficial. HVs described making referrals to food banks, charities, children’s centres, and housing support via Shelter. These were also identified as safe places and escapes for families, providing numerous health benefits. However, the pandemic stopped usual access, especially during lockdowns:

*“…we don’t have any power to change where they’re living…we don’t have that authority…it’s in the hands of the housing services or who’s working with that family, but you know, before COVID, at least we could provide them with alternative places, provide them with safe places that they could go to—children centres, activity groups, you know where, at least, that they can have that safe space where their children can play, where they can explore, and you know it would be an escape from their surroundings…because…being in these places would be detrimental to anybody’s mental health…it’s not healthy either…it’s quite small, it’s cold, it’s damp so, at least, we were able to provide…these safe places in the community where…their children could play and safely and learn these new skills and even be exposed to other children because they wouldn’t [or] might not have that [at] home,…which is important for them to develop…”* (HV4)

All groups commented on how repeatedly moving in/out of borough (i.e., transientness) impacted families’ social support on the community level and awareness of services in that area and how *“…it’s just really important to just not expect people to be able to just find these services, particularly now that there’s so many barriers in the way to doing that(NP1)*.” Some policies in TA included strict no-guest policies creating challenges in co-parenting and receiving support from family or friends. Other reasons parents experienced a lack of social support included not wanting visitors because there was no room and/or they were embarrassed with where they lived, which had a knock-on effect on parental mental health.

*"There’s low self-esteem for [the] parent. They are not happy. They’re not comfortable…with what they’ve got. They’re not able to mix with other parents…even when they take their children to school because they’re not proud of where they are living, so it has a significant impact, not on the child alone, [but] then even on the parent…” *(HV2)

Professionals emphasised that the pandemic had introduced new barriers by reducing their community presence. Many HVs, HPs, and NPs relied on community outreach to reach the most vulnerable U5TA who might otherwise be invisible and not have digital access, which was also a systems-level barrier. 

*“We probably aren’t reaching as many families as we could be because we don’t have a community presence at the moment with everything shift[ing] into digital services, and also not all families are tech savvy…we may do our initial contacts with our clients, but our main way of communicating with them moving forward to deal with their case would be via email and telephone, and it’s not very often that you will get good correspondence back and because it’s just not the way that they are used to dealing with things.”* (NP2)

Unawareness of their services prevented U5TA and their families from getting “*in the door”*(HP2) and *“to the service…*”(NP1), so professionals were dependent on referrals from other services, including charities. This became a more significant barrier during the pandemic when families, who usually did walk-ins, were referred, but the digital shift and poverty, especially language, lack of Wi-Fi, phone credit, etc., were also barriers. In contrast, LA1 said her clients were doing well despite COVID but associated this with her going the extra mile and ensuring that families could access health services.

#### 3.2.3. Systems Level

U5TA and their families experienced numerous top-down barriers, influencing their ability to access health services and optimise health outcomes. Moreover, professionals faced systemic challenges when providing their services, including discontinuity of care, low housing availability, virtual assessments, lack of outreach models, and heavy workloads. Many systems-level barriers were interrelated or precipitated barriers on the levels mentioned above. 

##### I. GP Registration, Service Closure, and High Demand

Difficulties in registering with a GP were frequently cited issues since GP surgeries are the gatekeepers to other health services in England. These difficulties stemmed from U5TA not having a fixed address, deregistering and registering with a new GP after moving out of borough, and language barriers because most GP appointment systems and registration intakes are in English only, as well as online creating digital inaccessibility. U5TA with NRPF status had even more difficulties accessing GP services, with LA1 commenting that they often wrote letters to the GP to try and facilitate access for the families. Professionals reported that such problems led to many missed immunisations, lack of continuity of care, records not being transferred efficiently and disengagement with health services. 

High demand and heavy workloads were commonly cited barriers. According to HP2, HVs were understaffed and did not have enough time to conduct home visits, which was the most critical environment to evaluate the child in versus the “sanitised view” in the clinic. Another HP said that therapists were understaffed and overworked, causing anywhere from a six-month to two-year waiting list for speech and language therapy. 

All professionals reported that COVID had exacerbated the difficulties families experienced accessing health services, including GP, health visitor, and dentist appointments, and most worried that, as a result, U5TA were behind in their mandated early years reviews/checks [38].

*“…when I’m thinking about it, it [COVID] has made a huge impact…I’ve got lots of families who want to [go to] dentists, and there’s specialist dentist service, but they can’t get to them. Health visitors aren’t going out…There’s a lot that’s changed for COVID because if we’ve all shut down, although we didn’t, this is what I don’t get, I still carried on—I did a lot of stuff with video, admittedly for a while, but we opened clinics up again. Not quite sure why everyone else hasn’t…even that GPs and things like, what’s happening? So, I do worry…they haven’t got the same access they would have before COVID because there just isn’t the appointments that were, everyone’s really behind.”* (HP4)

During COVID, the usual access channels for families to see the HVs, such as children’s centres, were reportedly not open, which meant that HVs had to reach out to those families proactively.

*“Because I think before…clients that I work with, quite often see the same health visitor like every week or something at the Children’s centre because that’s where he goes, so they just happened to kind of see them. But now it’s kind of relying on that client or that health visitor to go out of their way to have a call or visit, which obviously the health visitor has just to do to the best ability, but they have to see so many different families that maybe families those that they need support but who don’t know how to reach out for it.”* (NP1)

##### II. Continuity of Care and Transientness

HVs discussed the difficulties of knowing if a child had moved out of or into their area, which could only be determined once the family had registered with a different GP surgery. The lack of a centralised tracking system led to poor communication between clinicians and HVs about who is responsible for delivering care.

*“So they may be in the area, and we may not even know about them. But some of the barriers are also around whose clients are they because they [are] really registered to another local authority and therefore, one could argue that they are the responsibility of the health visitors in that area. However, while living temporarily in our area, I think geographically, we hold an element of responsibility, but there’s a barrier in communication between I think with some of the clinicians may experience, barriers in terms of communication around notification of movements into and out of the borough in a timely manner. For that really hinders developing…implementing effective care packages to…meet the individualised needs of those families.” *(HV3)

More specifically, HVs and HPs discussed how the transientness of families meant that therapy sessions stopped and had to be restarted whenever a family moved, which had a cumulative effect on their continuity of care. HP3 highlighted that the constant moving and lack of continued care was often distressing for children, particularly those with autism, as they struggle with change. 

##### III. Digital Inaccessibility and Virtual Appointments

Around half of the professionals reported digital inaccessibility as a barrier to accessing services, particularly acute during the pandemic when many services shifted to the digital realm instead of face-to-face/in-person [39]. Professionals gave various examples of how this shift affected many U5TA families who (a) did not have internet access (as this was not automatically provided to families living in TA) and/or a device to access the internet, (b) were not technologically savvy, and/or (c) were not comfortable with writing emails/answering the phone when English was not their first language. 

Professionals believed that many parents could not afford computers or tablets, making it difficult to schedule and access online appointments or only having one phone, making it challenging to tackle concurrent issues such as housing and benefits.

*“I think we all assumed that most families did [have internet access], but it’s still something that, yeah, not all have, especially if they’re in temporary accommodation. I think, you know, things aren’t always set up, and because they don’t know how long they’re going to be there…it’s something we’re still battling with and again.”* (HP3)

Some health professionals discussed that if families had reliable internet access, had a competent grasp of English, and did not want to come into the clinic/hospital, they could use online therapy videos provided by HPs. 

HVs discussed the challenge of virtually evaluating skills (e.g., developmental milestones) and relying on the parent’s testimony, especially during the early days of the pandemic when appointments were by phone.

*“…the technology as well, it’s not 100% sometimes, it can cause like glitches in the assessment. It might not work, or they can’t hear you; they can’t hear me. I think it’s better now, now that we have video, it’s made things better, the assessment better. As opposed to the beginning of lockdown when it was just on the telephone, which is more difficult because you can’t see anything, so all you can take is what the parent has told you at that point.”* (HV4)

However, professionals did have a “failsafe” to see U5TA in person if they were particularly concerned.

*“But then again, we always have failsafe, so we do have options to bring them into clinic. But then it’s not the ‘go to’ anymore, virtual is the ‘go to’ now, so I have found it challenging.”* (HV4)

NP1 noted that whilst many parents were not comfortable on Zoom, professionals also had to adapt to the new technology; thus, this was an issue for both parties. 

##### IV. Housing Availability

Housing availability was identified as a systems-level barrier, particularly in cities. According to professionals, some families were given housing far away from health services, which meant that both geographic distance and transportation costs to access health services became barriers. Although families were offered accommodation outside London, which were less polluted and more spacious, many did not want to move because of affordability, culture, fear of moving away from their support networks of family and friends, and fear of the “unknown”. 

*“…some families, they live in their boxes because they sometimes tell them, oh, you’re going to be here for three months, six months, and they’re waiting to be moved forever, and you’ve got families that they’ve been living in accommodation for ages, some of them up to eight [years] you know so they’re just sitting there waiting. Obviously,…the demand to live in London is more…” *(HV7)

LA3 attributed the housing shortage to boroughs bidding against each other for affordable housing and not enough to meet the demand. Some professionals mentioned contacting social care on behalf of families to ask the social care team to help with a housing placement when they were at risk of homelessness and were told that they had to wait until the last day of eviction before contacting them for help, which was too late for their families, i.e., they were made homeless. Many specialist HVs emphasised that U5TA with SEND waited extensive periods for suitable and accessible housing.

*“You receive an email from new housing saying there’s no houses out there for this family at the moment. There’s nothing. He may have to wait a year even being on emergency [list].” *(HV1)

Professionals discussed how the lack of stable housing triggered a top-down escalation of issues starting from housing which then impacts the child:

*“…I think that the housing, in a way, it can impact it’s like a chain reaction. It can impact loads of other parts of the family’s life. Most about primarily their health basically poor housing can equal to having poor health because it limits the access to health services if there are not services that are kind of in tune to picking up those families that are kind of transients basically.”* (HV4)

NPs acted as facilitators between families and GP surgeries while assisting with housing applications and benefits. Examples of teamwork among professional groups were highlighted as a way to get housing for priority need families (e.g., single-parent household and children with SEND), which included writing letters to LBN’s housing office to challenge inadequate housing on a policy level to improve the TA living conditions. 

##### V. Policies and Lack of Outreach Models

Although professionals were knowledgeable about some local and national policies in place that apply to families experiencing homelessness, most were not familiar with the significant policies that currently impacted this vulnerable group at the time of the interview (e.g., Homeless Reduction Act) or did not think there were any policies specific to TA or homelessness and under 5s.

Some were dismissive regarding the impact of existing policies on families experiencing homelessness: *“…if there is, then the policy is not working, and it’s not effective, because as far as I know, with or without a policy, [it] is still the same(HV5*).” Those aware of existing policies expressed that they were vague and lacked accountability for the suitability of TA and child safety. Reduction in government funding for schemes such as Sure Start and oral health programmes [40] plus service cuts due to COVID were also highlighted examples of how government policy directly impacted U5TA.

An HP and LA discussed the lack of outreach models (i.e., a proactive engagement approach focused on those least likely to access services) tailored to U5TA and their families, which only heightened their invisibility factor. Outreach models for U5TA were compared to those tailored to rough sleepers.

*“So, I think in terms of accessing, I think it is harder when population groups are a bit more dispersed …in the sense of I’m comparing it with rough sleeping and you can have more, there’s more outreach models that you’ll know certain hostels that you can work closely with. There might be outreach street workers. I’m not saying it in any way, this makes this easier, but you’ve got greater strategies. I think, with families in temporary accommodation, that can be a bit harder. I think they’re not necessarily as visible in terms of being seen as homeless.” *(LA3)

## 4. Discussion

To the best of our knowledge, this is the first qualitative study conducted in England exploring how various cross-sector professionals perceived the impacts of living in TA and the COVID-19 pandemic on U5TA’s healthcare service access and health outcomes. In addition, this study elucidated professionals’ own experiences and challenges in delivering services to this specific population. This study was conducted in one of England’s most linguistically, ethnically, and culturally diverse areas [41]. Various professionals, who worked with U5TA families in the London Borough of Newham, described multiple interrelated factors that were generally felt to be barriers to service access and to the parent/carer’s ability to support their child(ren) to optimal health and developmental outcomes. They largely agreed that barriers experienced by U5TA families included poor parental/carer mental health; unsuitable housing; transient circumstances; lack of social support or community; mistrust of mainstream services; and NRPF status, financial insecurity, food insecurity, and loss of informal jobs. Few enablers (e.g., children’s centres and charities as safe spaces) were reported but were less accessible to U5TA during lockdowns or frequent national guideline changes. Professionals reported mitigating some barriers using good communication skills and developing trusting relations. Facilitators were also important through the community or, in some instances, the professional [19]. 

The pandemic amplified health inequalities and inequities, disproportionately affecting the lives of U5TA and the ability of professionals to deliver quality care to U5TA across all five domains of access [18,19]. According to professionals, in-person services were reduced as the pandemic shifted to remote delivery of health and social care services, which exacerbated pre-existing barriers. Consequently, differential impacts of digital poverty, language discordance, and inability to register and track U5TA rendered them invisible to these services. All professional groups reported closures of local children’s centres during lockdown. However, these services remained open throughout the pandemic, albeit with reduced services and some safety restrictions regulating who could attend (e.g., vulnerable children), thereby suggesting a potential communication gap among different services and one of the reasons why attendance fell during this time [42,43].

While a strength of this study is the cross-sector perspective, findings from previous research support the themes identified in this research, strengthening and validating these findings [2,14,33,34]. For example, professionals’ perceived barriers to delivering care to UT5A, paired with the increased workload and demand on HVs in particular, were comparable to another study conducted on health visiting in England during the pandemic—38% of respondents (n = 253/663) who had their caseload increase by 50% or more, and 41% lost staff on their teams due to redeployment [44]. Both studies indicate that hiring additional staff is essential to meet the U5TA needs and prevent staff burnout and dissatisfaction.

In the housing environment, severe Housing Health and Safety Rating System (HHSRS) [45] hazards and lack of safety provisions (i.e., baby/childproofing) for stairs and windows found in this study were similarly discussed in a recent study looking at the suitability of TA, including the use of fire escapes for building entry [14]. In the ENFAMS (Enfants et Familles sans Logement) survey of families living in emergency centres, long-term rehabilitation centres, social hostels, and centres for asylum seekers in France, anaemia was positively associated with child food insecurity, no cooking facilities, and monthly household income in the 0.5–5 years stratified age group [46]. The TA environment was perceived unsafe for UT5A (e.g., sleeping arrangements), and while working on-site, some professionals uncovered that current policies were not being followed or strictly enforced. This included the Licensing and Management of Houses in Multiple Occupation and Other Houses 2006 regulations on the bathroom-to-tenant ratio in TA for multiple-sharing households (i.e., no less than one bathroom/bath or shower/lavatory for every five people sharing) [47,48,49]. This evidence indicates that TA was overcrowded and increased the risk of infection, especially during the pandemic when it was difficult, if not impossible at times, to follow national social distancing guidelines [50].

Similar to previous research, professionals commonly reported poorer health outcomes (e.g., respiratory infections, anaemia, and asthma) and poor access to health services (e.g., vaccine delay, continuity of care) among children experiencing homelessness [2]. Moreover, one recommendation professionals strongly emphasised was that families must be able to access free and safe spaces for health services, which could also provide opportunities to leave their surroundings and interact with other children to develop critical physical, cognitive, and social-emotional skills through play and engagement. A previous qualitative study of families with lived experience also identified space obstacles and a need for children’s activities, as well as the top three barriers discussed by professionals in this study: transportation, employment, and child care [51]. Likewise, those deprivations, barriers to housing and services, and indoor living environments have been shown to be important predictors of child mortality [52]. The combination of poorer outcomes and adverse childhood experiences (ACEs), some of which occur before the child is born, have short- and long-term implications for the child and adult lives of U5TA, including low birth weight and pre-term births, chronic health conditions, and mortality [52,53,54]. 

Poor parental mental health was also a dominant theme discussed by professionals, which was perceived to be caused by a plethora of interacting factors, including homelessness status or transientness, poor quality TA, no social support network, and many competing priorities, which then affected the child(ren)’s health and wellbeing. This is supported by quantitative findings that living in TA increases the odds of poor parental mental health [34]. The Millennium Cohort Study also found an association between moving more frequently and poor self-rated health [55]. For comparison, in the US, mothers with a history of homelessness had higher adjusted odds of fair or poor health and depressive symptoms compared to consistently housed mothers [56]. The ENFAMS study concluded that children growing up and experiencing homelessness have psychological difficulties, which can risk long-term poor mental health and educational outcomes [57]. However, unlike this study, these quantitative studies lacked qualitative insights from cross-sector professionals.

### 4.1. Strengths and Limitations

This was the first study to obtain a cross-sector professional point of view with a specific focus on U5TA. A strength of this study was the qualitative insight, including in-depth front-line accounts with a range of professionals. This study also explored professionals’ perspectives on the impact of COVID on health outcomes and their experiences working with U5TA and their families, including the services they were able to provide during this time, areas that have yet to receive much attention in the literature [58]. An additional strength of this study was the focus on the barriers experienced by those with SEND in TA [2], a sub-group often missed. These interviews demonstrated the need for critical evaluation of current policies that impact U5TA, including children’s rights, housing, and healthcare provision. Including key professionals across sectors allowed for systems mapping to better understand the housing crisis complexities in which public health challenges emerge [59]. 

There are also several limitations of this study. Potential bias was possible, as in many qualitative studies, but was minimised by using rigorous and replicable methods. Although qualitative research is said to be limited in its ability to quantify impact [60], the previously mentioned studies had similar findings, thus supporting this study’s results. Furthermore, by examining the presence or absence of themes/sub-themes among professionals per group, this study provided data on the visibility of specific issues (e.g., unsuitable housing, immigration) (Appendix A). Another limitation was recruitment difficulties due to the unpredictability of lockdowns, increasing the workload [44] for professionals and possibly explaining the lack of representation from the LBN Housing department; therefore, that was a missing piece of the analysis and would have added value in the systems mapping process and future research. While every study has its limitations, the findings may still be generalisable to other populations in England, given the high prevalence of child homelessness in TA and the fact that many of the professionals had also worked outside of LBN with U5TA.

### 4.2. Implications and Future Research

From the professional perspective, one of the significant implications was how many barriers were interrelated, with a ripple effect often causing more barriers. In addition, the pandemic was found to exacerbate existing barriers, demonstrating how U5TA are vulnerable to the adverse effects of the pandemic and have the potential to be left more vulnerable to the ongoing impact of the post-pandemic recession. To address the multiple, interrelated barriers, professionals recommended creating opportunities for parents/carers to help alleviate some of the burdens related to living in TA (e.g., financial and food insecurity, NRPF status). However, there was also recognition that national policy would still be needed to address systems-level barriers that are the root causes of health inequalities [61,62].

The transientness of the families was found to be a consistent theme across all barriers impacting not only healthcare access but also a significant determinant of health for U5TA and their families. This highlights that these barriers require cross-sector, progressive actions tailored to their specific needs to address their heightened vulnerability. These qualitative insights also further support housing as a socio-political determinant of health inequality [63] and a driver of significant inequity [33], which has policy implications on the local, national, and global levels. Furthermore, professionals were asked for recommendations in addressing these challenges, including developing policy, tailored outreach models, and emergency preparedness strategies for future pandemics. As part of a multi-phase project, policy-level implications, in combination with the professionals’ recommendations, will be published in a forthcoming manuscript, including a TA Standards Framework.

This study reported local-level findings, so further research is needed to compare its results with evidence from other local authorities in London and, more broadly, England. The next steps would be to continue this body of work in other UK regions and implement the same study design in countries with comparable rates of homelessness and similar geographic distributions of homelessness (i.e., higher prevalence in cities), such as the United States. The purpose would be to compare systems, support services, and policies across different countries to learn how these differences help or hinder families living in TA. 

As shown in this study, qualitative research needs to be adaptable and flexible to the challenges introduced by the pandemic. The ability to conduct quality interviews online increases the likelihood of scheduling interviewees and even globalises research by allowing for interviews with leading field experts and experts by experience from across the world [14]. 

## 5. Conclusions

According to various key professionals, the U5TA population, regardless of the COVID-19 pandemic, experiences significant barriers and limited facilitators to optimising health outcomes and accessing healthcare services, which makes them particularly vulnerable and less visible. Therefore, innovative cross-sector strategies and integrated care systems (i.e., housing and health) are needed to address these barriers more broadly. The COVID-19 pandemic amplified some of these barriers via existing and/or new mechanisms (e.g., digital exclusion) and reduced vital facilitators, including co-located community services. There is now a generation of U5TA who have experienced potentially severe effects on their health and development, such as delays in potty training and/or regression back into nappies, no longer self-feeding, challenging behaviour, and reaching milestones, which likely have life-long consequences. Ultimately, tailored and co-produced strategies to ’repair’ the damage experienced by this generation are urgently needed whilst ensuring future generations, who find themselves in TA, do not experience the same barriers.

## Figures and Tables

**Figure 1 ijerph-20-01300-f001:**
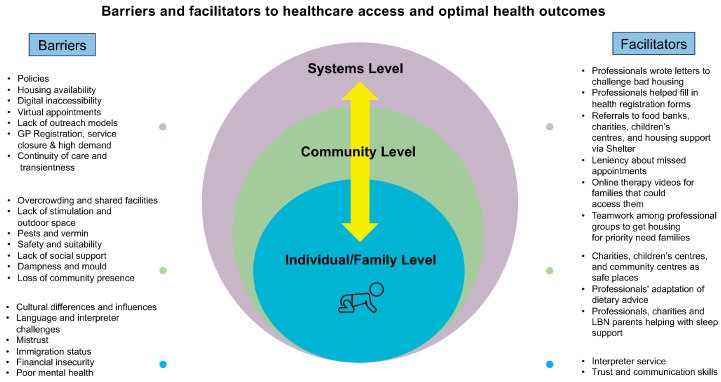
Barriers and facilitators to healthcare access and optimal health outcomes.

**Table 1 ijerph-20-01300-t001:** Professional Characteristics.

Professional Group	Number of Professionals n (%)
Health Visitors	7 (44)
Health Professionals	4 (25)
Non-profit Sector	2 (13)
Local Authority	3 (19)
**Age Group**	**n (%)**
25–29	3 (19)
30–34	1 (6)
35–39	1 (6)
40–44	2 (13)
45–49	4 (25)
50–54	3 (19)
55–59	1 (6)
60–64	1 (6)
**Gender**	**n (%)**
Male	2 (13)
Female	14 (88)
**Ethnic Group**	**n (%)**
White (British and/or Any other White British background)	7 (44)
Black (African or Caribbean)	5 (31)
Asian or Asian British: Bangladeshi	1 (6)
Mixed Ethnic background	3 (19)
**How Long Professionals Have Been in Current Role** **(Time Spans)**	**n (%)**
=<6 months	3 (19)
7 month–1 year	2 (13)
1.1–2.5 years	5 (31)
2.6–5 years	1 (6)
5.1–10 years	4 (25)
>10 years	1 (6)
**Total**	**16 (100%)**

Note: Percentages are rounded to the nearest whole number, so percentages may not add up to exactly 100%.

## Data Availability

Additional tables and matrices can be made available upon special request, although they may not be republished elsewhere.

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
