# Peer review of "How Does Living in Temporary Accommodation and the COVID-19 Pandemic Impact under 5s’ Healthcare Access and Health Outcomes? A Qualitative Study of Key Professionals in a Socially and Ethnically Diverse and Deprived Area of London"

_ijerph, 2023, doi:10.3390/ijerph20021300_

Round 1

Reviewer 1 Report

The article is well written (as a non-native speaker who learns English all over the life I enjoyed the language indeed) and deals with important social issues. Its style is appropriate also, but the paper needs some technical corrections and addresses some questions given below.

First, the dot after every literature source cited in the square brackets should be placed AFTER these brackets at the end of the sentence. Not before, as it is done throughout the paper now. E.g. line 49 now: …such as rough sleepers.[3] Should be: such as rough sleepers [3]. Line 59 now: …ment, relationships).[3,6,7] should be …ment, relationships) [3,6,7].

This issue should be checked and corrected all over the text.

The second. Check all abbreviations all over the text please. For example, despite usage of abbreviation TA seems logical, it is not introduced, explained in the text, e.g. at the line 60, where this abbreviation is used for the first time.

Line 53 - was affected, by these measures, - comma after ‘affected’ seems redundant

Line 70: from 2019(pre-pandemic) – space is missed

Line 117-119, you wrote: Potential participants were emailed and sent a participant information sheet, and if interested, were asked to reply via email with any questions and/or to schedule the interview. – It is not very clear at this point if you used this email answers for assessment (as far as I understand from the further description – not) instead of the personal interview. As is social studies face-to-face and indirect contacts can lead to different results of the study (affecting honesty of the interviewee), the way all results were obtained needs clear expression all over the text.

Line 129: “to be effective24,” should be replaced by “to be effective [24],”

Line 170-171: Sixteen interviews were conducted (median time taken: 28 minutes; range: 14-41 minutes). This sentence should be incorporated into the Method.

Lines 184 and 186 – spaces are missed before (NRPF) and (SEND) respectively.

Lines 206-207: Professionals mentioned fewer facilitators, including good communication skills and building trusting relations. This sentence seems senseless or, better is not well corresponding with the previous one, it is difficult to understand what was mentioned here.

Lines 507-512 look like a quote of NP2, but they are given as a authors' text, not like a quotation

Lines 754 and 781: Numbers of paragraphs 6.5.1. (Strengths and Limitations) and 6.5.2. (Implications and Future Research) do not correspond with the number of Discussion chapter, which is 4.

Line 760 now: “in the literature.58” Should be: “in the literature [58].”

Lines 769 and 774 spaces are missed before citation sources.

Author Response

The article is well written (as a non-native speaker who learns English all over the life I enjoyed the language indeed) and deals with important social issues. Its style is appropriate also, but the paper needs some technical corrections and addresses some questions given below.

Thank you for your feedback and taking the time to read our manuscript. We are pleased that you found the language and style accessible and appropriate.

First, the dot after every literature source cited in the square brackets should be placed AFTER these brackets at the end of the sentence. Not before, as it is done throughout the paper now. E.g. line 49 now: …such as rough sleepers.[3] Should be: …such as rough sleepers [3]. Line 59 now: …ment, relationships).[3,6,7] should be …ment, relationships) [3,6,7].

This issue should be checked and corrected all over the text.

We have checked and corrected this.

The second. Check all abbreviations all over the text please. For example, despite usage of abbreviation TA seems logical, it is not introduced, explained in the text, e.g. at the line 60, where this abbreviation is used for the first time.

We have checked and addressed this.

Line 53 - was affected, by these measures, - comma after ‘affected’ seems redundant

We have removed the comma.

Line 70: from 2019(pre-pandemic) – space is missed

We added the space.

Line 117-119, you wrote: Potential participants were emailed and sent a participant information sheet, and if interested, were asked to reply via email with any questions and/or to schedule the interview. – It is not very clear at this point if you used this email answers for assessment (as far as I understand from the further description – not) instead of the personal interview. As is social studies face-to-face and indirect contacts can lead to different results of the study (affecting honesty of the interviewee), the way all results were obtained needs clear expression all over the text.

Thank you for your feedback. We had cut some of this material out because the paper became very long. We’ve added back some minor details that hopefully clarify the methods. Potential participants were asked to express their interest in taking part via email.

Line 129: “to be effective24,” should be replaced by “to be effective [24],”

We have corrected this.

Line 170-171: Sixteen interviews were conducted (median time taken: 28 minutes; range: 14-41 minutes). This sentence should be incorporated into the Method.

Thank you for your feedback. We consider this an output which is why it is in the Results, but we think it could also be in the Methods, so it’s a matter of style preference and both work.

Lines 184 and 186 – spaces are missed before (NRPF) and (SEND) respectively.

We have corrected this.

Lines 206-207: Professionals mentioned fewer facilitators, including good communication skills and building trusting relations. This sentence seems senseless or, better is not well corresponding with the previous one, it is difficult to understand what was mentioned here.

We tried to clarify this sentence: “In contrast, professionals only mentioned a few facilitators, including good communication skills and building trusting relations.”

Lines 507-512 look like a quote of NP2, but they are given as a authors' text, not like a quotation

The quote is from (HV2) which is at the end of the quote.

Lines 754 and 781: Numbers of paragraphs 6.5.1. (Strengths and Limitations) and 6.5.2. (Implications and Future Research) do not correspond with the number of Discussion chapter, which is 4.

We have corrected this.

Line 760 now: “in the literature.58” Should be: “in the literature [58].”

We have corrected this.

Lines 769 and 774 spaces are missed before citation sources.

We have corrected this.

Reviewer 2 Report

This is an overall well-written manuscript by the authors. The authors conducted a qualitative study with 16 online interviews. The objective is to explore professionals’ perspectives of barriers and facilitators for children <5 years living in temporary accommodation to accessing health care services. The authors also aim to elicit the professionals’ experience when providing services to this population before and during the COVID-19 pandemic. The study methods and inclusion/exclusion criteria are clearly specified. The results are clearly presented at different levels and discussed in detail. The results and conclusion support the objectives and they are relevant to the journal’s aims. The limitations are acknowledged and discussed in a reasonable manner for future research. This manuscript provides useful information to guide future research and health care service for the unique population of children <5 years living in temporary accommodation. The findings are important to its relevant field and scientific community. I have the following minor comments that I hope the authors could address:

1.     Table 1 is not professional to report categorical results with counts and percentages. The total numbers do not need to be repeated and the total percentage is not necessary if the table could be revised to standard tables with an overall header and reporting categories in n and %. Especially with the rounding and summing exceeding 100%. Please revise according to common practice in scientific publications and tables.

2.     More on Table 1, it would be preferred to include more socio-demographic characteristics variables if they are collected. This helps better understand the diversity and background of the study participants. Also, results of continuous characteristic variables, such as age and time, could be reported in both categorical and continuous summaries using median/mean and range/quartiles/standard deviation.

Author Response

This is an overall well-written manuscript by the authors. The authors conducted a qualitative study with 16 online interviews. The objective is to explore professionals’ perspectives of barriers and facilitators for children <5 years living in temporary accommodation to accessing health care services. The authors also aim to elicit the professionals’ experience when providing services to this population before and during the COVID-19 pandemic. The study methods and inclusion/exclusion criteria are clearly specified. The results are clearly presented at different levels and discussed in detail. The results and conclusion support the objectives and they are relevant to the journal’s aims. The limitations are acknowledged and discussed in a reasonable manner for future research. This manuscript provides useful information to guide future research and health care service for the unique population of children <5 years living in temporary accommodation. The findings are important to its relevant field and scientific community. I have the following minor comments that I hope the authors could address:

Thank you for your feedback and for taking the time to read our manuscript. We greatly appreciate that it has been well received and that you believe it will make an important contribution to the field and scientific community.

  1. Table 1 is not professional to report categorical results with counts and percentages. The total numbers do not need to be repeated and the total percentage is not necessary if the table could be revised to standard tables with an overall header and reporting categories in n and %. Especially with the rounding and summing exceeding 100%. Please revise according to common practice in scientific publications and tables.

We have revised the table, so there is one total at the bottom, and n (%) are in the same column. We also rounded the % to the nearest whole number.

  1. More on Table 1, it would be preferred to include more socio-demographic characteristics variables if they are collected. This helps better understand the diversity and background of the study participants. Also, results of continuous characteristic variables, such as age and time, could be reported in both categorical and continuous summaries using median/mean and range/quartiles/standard deviation.

We only reported on socio-demographics that were non-identifiable. The others would have given away too much information on our professionals. Age was collected as a categorical variable, and we already have the range of time in the text at line 181 and have now added the median and standard deviation.
